# Representing Joint Hierarchies with Box Embeddings

**Dhruvesh Patel** *                                    DHRUVESHPATE@CS.UMASS.EDU
**Shib Sankar Dasgupta** *                                 SSDASGUPTA@CS.UMASS.EDU
**Micheal Boratko**                                          MBORATKO@CS.UMASS.EDU
**Xiang Li**                                                     XIANGL@CS.UMASS.EDU
**Luke Vilnis**                                                    LUKE@CS.UMASS.EDU
**Andrew McCallum**                                         MCCALLUM@CS.UMASS.EDU
*University of Massachusetts Amherst*

## Abstract

Learning representations for hierarchical and multi-relational knowledge has emerged as an active area of research. Box Embeddings [Vilnis et al., 2018, Li et al., 2019] represent concepts with hyperrectangles in $n$-dimensional space and are shown to be capable of modeling tree-like structures efficiently by training on a large subset of the transitive closure of the WordNet hypernym graph. In this work, we evaluate the capability of box embeddings to learn the transitive closure of a tree-like hierarchical relation graph with far fewer edges from the transitive closure. Box embeddings are not restricted to tree-like structures, however, and we demonstrate this by modeling the WordNet meronym graph, where nodes may have multiple parents. We further propose a method for modeling multiple relations jointly in a single embedding space using box embeddings. In all cases, our proposed method outperforms or is at par with all other embedding methods.

## 1. Introduction

Hierarchical relations, particularly hypernymy, are inherently present in natural language and useful in many common tasks. For example, in SNLI the sentence "A soccer game with multiple males playing" can be understood to entail "Some men are playing a sport" via the hypernym relation between "soccer" and "sport". Question answering and language modeling, in general, can benefit from understanding these relationships, as a word can often be replaced by its hypernym and still yield a valid sentence or provide supporting evidence to answer a question. Vector embedding methods, where an entity is associated with a vector in $\mathbb{R}^n$, often use symmetric measures such as the dot product between these representations which cannot capture inherently asymmetric relations such as hypernymy.

In order to solve this problem, [Vendrov et al., 2016] introduced an asymmetric measure on vectors in the form of entailment cones in $\mathbb{R}_+^n$. [Vilnis et al., 2018, Li et al., 2019] generalized this notion to propose box embeddings which are hyperrectangles in $\mathbb{R}^n$. We build upon this work by proposing a new regularization loss for the box embedding model. This provides a weak scale on the embedding space resulting a significant performance improvement.

In this work, we explore the extent to which the edges from the transitive closure are necessary to model the tree-like large tree-like graphs, e.g, WordNet's hypernymy. We

---

∗. Equal Contribution

further explore the meronymy relation from WordNet, which differs qualitatively from the hypernymy graph in that it is not as tree-like, comprised of many connected components where some nodes have multiple parents while others have no parents. We not only achieve state-of-the-art performance with our box embedding based method over other baselines when 10, 25, and 50% of the transitive closure edges are supplied during training but, most importantly, we outperform other methods by a significant margin when the training is being restricted to the transitive reduction only (0% transitive closure edges).

Finally, we introduce a method of modeling these two relations in the same space by introducing two representations for each entity. Our method is not specific to box embeddings, but rather can be thought of as a graph augmentation procedure which is general enough to allow for modeling with all existing baselines. In this multi-relational embedding settings, we also compare against knowledge base completion baselines and demonstrate that box embeddings can outperform TransE [Bordes et al., 2013] and ComplEx [Trouillon et al., 2016] in this setting as well[*].

## 2. Related Work

Using distributed representations to capture semantic relationships amongst words/concepts has had a long history [Mikolov et al., 2013, Pennington et al., 2014]. However, due to the use of a symmetric distance metric, these models are incapable of representing the asymmetric relations like *hypernymy*. Recent works that address this issue can be broadly categorized into two approaches: (1) using point representations combined with an asymmetric distance measure, and (2) using region-based representations with a probabilistic interpretation.

While Nickel and Kiela [2017], Chamberlain et al. [2017] use points in n-dimensional hyperbolic space (Poincaré ball) to model tree-structured data, Vendrov et al. [2016] uses the asymmetric measure of the reverse product order on points in Euclidean space to perform the same task. Using the reverse product order amounts to associating with each entity a Euclidean cone with apex $x$, which is the set $\{y : \bigwedge_{i=1}^{n} y_i \geq x_i\}$. Ganea et al. [2018] extend the hyperbolic point embeddings to entailment cones in hyperbolic space.

Another line of work uses distributions over points in space to learn representations for hierarchical data[Vilnis and McCallum, 2015, Muzellec and Cuturi, 2018]. Lai and Hockenmaier [2017] provide a probabilistic interpretation of order embeddings [Vendrov et al., 2016] by using the negative exponential measure or, equivalently, representing concepts as a cone in $[0, 1]^n$. With this parameterization, the volume of a cone could be interpreted as the marginal (resp. joint) probability of the entity (resp. intersection) which it represents. This allowed for training and evaluation using conditional probabilities which mapped very naturally onto hypernym relations, eg. $P(\text{mammal}|\text{dog}) = 1$. However, representing entities probabilistic cones has a serious deficit – it cannot represent negative correlation, that the case when $P(X|Y) \geq P(X)$. Box embeddings, introduced in Vilnis et al. [2018], solves this issue by introducing another vector, effectively associating each entity with an $n$-dimensional hyper-rectangle. Their method has been shown to effectively model hypernymy by training on a large subset of the transitive closure of the WordNet hypernym graph. In this work, we propose a regularization loss to further improve the performance of box embeddings.

---

*. Source code and datasets used in the paper are available at `https://github.com/iesl/Boxes_for_Joint_hierarchy_AKBC_2020`

We also present a modelling technique that extends the previously discussed methods to jointly model multiple hierarchical relations.

## 3. Method

We first present box embeddings as they were introduced in [Vilnis et al., 2018], briefly touch on the "soft" volume calculation presented in [Li et al., 2019], and then discuss some learning adaptations present in our version of the model.

### 3.1 Representation

The box-space [Vilnis et al., 2018] is the set of all $n$-dimensional hyperrectangles. Formally, if $\mathbb{B} = \{[a, b] \mid a, b \in \mathbb{R}, \ b \geq a\}$ is the set of all closed intervals of $\mathbb{R}$, then an n-dimensional box-space is $\mathbb{B}^n$. Just as in vector embedding models where each entity is represented by a vector in the euclidean space, in box-space each entity $e$ is represented by a box $\alpha_e \in \mathbb{B}^n$ as shown in Figure 1. We parametrize a box using the minimum and maximum coordinates in each dimension, $(\alpha_m, \alpha_M)$, where $\alpha_m, \alpha_M \in \mathbb{R}^n$ and $\alpha_{m,i} \leq \alpha_{M,i}$ for each $i \in \{1, \ldots, n\}$. Box embeddings provide a natural way to embed a partial order using containment.

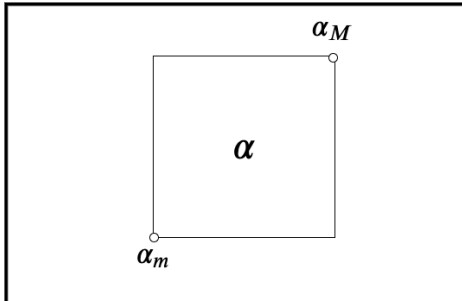

Figure 1: An element of box-space.

### 3.2 Probabilistic Interpretation

In addition to the natural containment operation, we can consider the *volume* of boxes. If we normalize the space to have volume 1 (by dividing by the smallest containing box, say) the boxes can be interpreted as a parameterization of a joint probability distribution over binary random variables, where the marginal probability of a given entity is given by the volume of its box. Intersections of boxes yield either another box or the empty set, and in either case, we can compute the volume of the intersection as

$$\text{Vol}(\alpha \cap \beta) := \prod \max(\min(\alpha_{M,i}, \beta_{M,i}) - \max(\alpha_{m,i}, \beta_{m,i}), 0). \tag{1}$$

We can interpret this volume as the joint probability for concepts $\alpha$ and $\beta$, and furthermore calculate (for example) [†]

$$\Pr(\alpha|\beta) = \frac{\text{Vol}(\alpha \cap \beta)}{\text{Vol}(\beta)}. \tag{2}$$

Vilnis et al. [2018] uses these conditional probabilities to model the hypernym graph of WordNet. For example, given an edge from MAN to PERSON (indicating that a man is a person) we first convert this to a conditional probability between binary random variables, $\Pr(\text{PERSON}|\text{MAN}) = 1$, which is then trained using binary cross-entropy loss and stochastic gradient descent. Figure 2 shows the corresponding boxes capturing this edge.

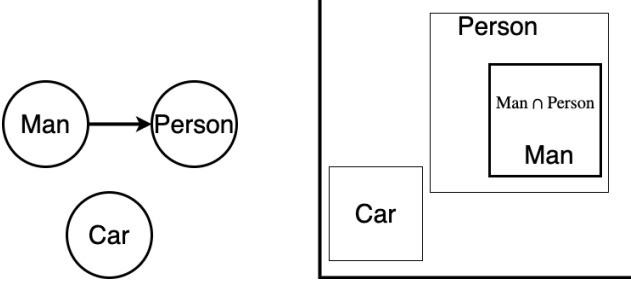

Figure 2: Formulation of conditional probability using intersection of boxes. Note that in this figure Man ∩ Person = Man.

### 3.3 Softplus Volume

Box embeddings that should overlap may become disjoint, either due to initialization or during training, and the currently described model has no training signal in this instance. Vilnis et al. [2018] made use of a surrogate loss function to handle this case, however Li et al. [2019] introduced a new method which replaces the ReLU from (1) with the softplus function, $\text{softplus}_t(x) = t \log(1 + e^{x/t})$, which smoothed the loss landscape and resulted in easier training. The volume calculation under this approximation becomes

$$\text{Vol}(\alpha \cap \beta) := \prod \text{softplus}_t(\min(\alpha_{M,i}, \beta_{M,i}) - \max(\alpha_{m,i}, \beta_{m,i})). \tag{3}$$

We note that (3) is always positive, which means all boxes "intersect" one another under this volume calculation. The temperature parameter $t$ can be tuned, and as $t \to 0$ we recover the original box model. We will exclusively use this soft volume calculation when computing box volumes, and in our experiments, we always select the temperature based on dev set performance.

### 3.4 Absence of explicit probability labels and use of regularization

Previous work on box embeddings [Vilnis et al., 2018] uses pre-computed unary marginals $\Pr(A)$ and pairwise conditionals $\Pr(A \mid B)$ as labels while training the box embedding

---

†. Note that when calculating conditional probabilities it is not necessary to normalize the volume of the space to size 1.

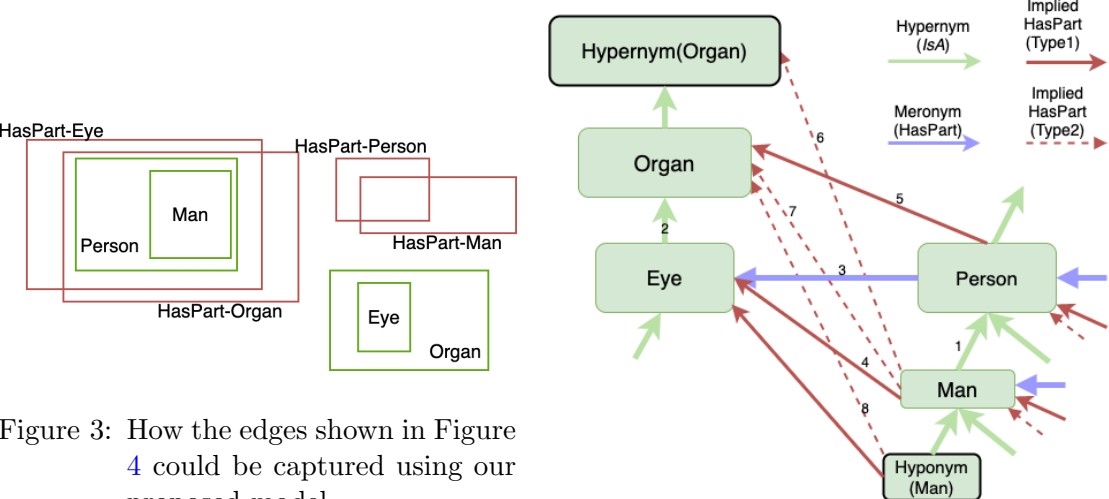

Figure 3: How the edges shown in Figure 4 could be captured using our proposed model.

Figure 4: Implied edges with the semantics of Has-Part when the hypernym and meronym are viewed as two transitive relations.

model. These marginal probabilities were calculated using the graph structure - leaves were given marginal probability equal to 1/#nodes, and parents were assigned marginal probability equal to the sum of their children plus 1/#nodes. How "tight" the parent boxes are to their children can be seen as a form of regularization. In general, setting marginal probabilities can help to set a scale for the overall model.

For our model, however, we avoid pre-computing marginal probabilities and simply use the conditional probability targets with binary cross-entropy loss and augment it with an additional regularization loss which can be written as

$$L = \sum_i^N -y_i \log p_i - (1 - y_i) \log(1 - p_i) + \sum_j^{N_e} \mathbb{I}_{\left[\text{Vol}(\alpha^{(j)}) > \tau\right]} \ \text{Vol}(\alpha^{(j)}) \ , \qquad (4)$$

$N$ being the total number of examples (including the random negatives), $N_e$ the total number of entities, $y_i$ the label, $p_i$ the conditional probability calculated using equation 2, and $\mathbb{I}$ the indicator function.

Without any global scale i.e., the marginal probabilities, we get poor training results, so we introduce this regularization measure by penalizing the size of boxes when they become greater than a fixed volume. This provides a weak scale to the boxes in the embedding space and results in significant performance improvement over unregularized training without the need to precompute unary marginals.

### 3.5 Modeling joint hierarchy

Inter-relational semantics implicit in multi-relational knowledge graphs can be used to learn coherent representations for the entities present in the graph. In particular, the relations

discussed in this work, i.e., the IsA and HasPart relations should obey the following rules.

$$\text{IsA}(a,b) \wedge \text{HasPart}(b,c) \Rightarrow \text{HasPart}(a,c) \tag{5}$$

$$\text{IsA}(a,b) \wedge \text{HasPart}(c,a) \Rightarrow \text{HasPart}(c,b) \tag{6}$$

For instance, every Man IsA Person and every Person HasPart Eye. Hence, every Man HasPart Eye. Similarly, Eye IsA Organ and Person HasPart Eye implies Person HasPart Organ. Figure 4 shows the subgraph describing this example. The red edges and the dotted red edges are implied HasPart edges which might be missing in the original knowledge graph. We show that boxes can model these two relations in a single space by using two boxes per node while preserving the semantics described above.

For each node $x$ we embed two boxes, one which represents "Things which are $x$" and one which represents "Things which have $x$ as a part". This is depicted in Figure 3, where the entity "Man" is represented by the green Man box and the red HasPart-Man box. The edges 3,4,5 and 7, in Figure 4 are captured by the inclusion of Man and Person boxes inside the HasPart-Eye and HasPart-Organ boxes, while the edges 1 and 2 are captured by the inclusion of Man inside the Person Box, and Eye inside the Organ box, respectively.

This form of modeling multiple relations using augmentation is not limited to the box embedding model. The main idea is to create a new graph with twice as many nodes and edges from the IsA hierarchy, with additional PartOf edges so as to make the semantics coherent. Any model capable of modeling a DAG can be evaluated on its ability to model this graph.

## 4. Experiments

In order to demonstrate the efficacy of the box embeddings model on hierarchical structured data, we evaluate on the hypernymy and meronymy relations of WordNet. Following the recent works of [Nickel and Kiela, 2017, Ganea et al., 2018], we train on the transitive reduction, using increasing amounts of edges from the transitive closure of those lexical relations. We pose the problem as a binary classification task on the unseen test edges which include the remaining edges from the transitive closure along with a fixed set of random negatives for the respective relations.

### 4.1 Datasets

1. Hypernymy (IsA)

   For hypernymy, we used the dataset from [Ganea et al., 2018]. The WordNet noun hypernymy dataset contains 82,114 nodes (omitting the root). The transitive reduction of the hypernymy hierarchy is used for training. Additionally, there are training datasets with 10%, 25%, and 50% of the transitive closure edges. For dev and test datasets, 10% of the edges (5% each) are sampled from the rest of the transitive closure. For each test and validation edge, a fixed set of negative samples of size 10 was generated.

|          | Transitive Closure | Transitive Reduction | Validation (pos/neg) | Test (pos/neg) |
|----------|--------------------|----------------------|----------------------|----------------|
| Hypernym | 84363              | 661127               | 28838/288380         | 28838/288380   |
| Meronym  | 9678               | 30333                | 5164/51640           | 5164/51640     |

Table 1: Details of the hypernymy and meronymy hierarchies. This training set corresponds to the 0% transitive closure setting.

2. Meronymy (HASPART)

   We created a meronomy dataset in the exact same way. Out of the 82,114 nodes in WordNet, only 11,235 of them have meronym relations associated with them. We note that this graph is not nearly as tree-like - there are 2,083 connected components, 2,860 nodes with no parents and, 1,013 nodes with multiple parents. Again, following [Ganea et al., 2018], we start with the transitive reduction and add 0%, 10%, 25%, and 50% of the transitive closure to the training data in order to observe the corresponding improvement in the performance of the model. Dev and test datasets were created in the same way as for hypernymy. Please refer to Table 1 for the details of the dataset.

3. Joint Hierarchy (IsA and HASPART)

   For the joint hierarchy, we start with the transitive reduction of meronymy and hypernymy and create a new graph as described in Section 3.5. Specifically, we add the implied HASPART edges (shown as the red arrows in Figure 4). The test data is comprised of a subset of the implied edges (shown as dotted red in Figure 4) which are of the form

$$\underbrace{\text{IsA} \circ \text{IsA} \cdots \text{IsA}}_{m \text{ times}} \circ \text{HASPART} \circ \underbrace{\text{IsA} \circ \text{IsA} \cdots \text{IsA}}_{n \text{ times}} = \text{HASPART}. \tag{7}$$

   The dev and test created in this manner have 94807 and 94806 positive edges respectively which include all HASPART edges as described in (7) for all $m, n \in \{1, 2\}$.

## 4.2 Baselines

We compared our model with the recent strong baseline models such as:

1. Order Embeddings [Vendrov et al., 2016]: As mentioned in the introduction, this method embeds entities as cones in $\mathbb{R}+^d$. Representationally, order embeddings are equivalent to box embeddings where the min coordinate is fixed to the origin, however, learning differs in that the objective function optimizes an order-violation penalty. (See [Vendrov et al., 2016] for more details.)

2. Poincaré Embeddings [Nickel and Kiela, 2017]: In this work, the authors learn embeddings in an $n$-dimensional Poincaré ball which naturally captures the tree-like structure via the constant negative curvature of hyperbolic geometry. This work uses

a symmetric distance function which is not ideal for asymmetric relational data, however. We use the same heuristics followed by [Ganea et al., 2018] to mitigate this problem.

3. Hyperbolic Entailment Cones [Ganea et al., 2018]: This work models hierarchical relations as partial orders defined using a family of nested geodesically convex cones in hyperbolic space.

4. TransE and ComplEx [Bordes et al., 2013, Trouillon et al., 2016]: The joint hierarchy can be also considered as multi-relational data with two relations. We use the most popular translational distance-based and matrix factorization based models, TransE and ComplEx respectively.

In order to show the effectiveness of the proposed regularisation, we include the performance of the vanilla box embeddings [Li et al., 2019] as well.

### 4.3 Training details

In all our experiments, we keep the embedding dimension as 10 for all the baseline models, while for our model we use embedding dimension as 5. This results in having 10 parameters per node and ensures a fair comparison. For the task of learning the joint hierarchy with multi-relation prediction models like TransE and ComplEx, we keep their embedding dimension as 20 to account for the doubling of parameters due to the introduction of two embeddings per node in Order embeddings, Poincaré embeddings, Hyperbolic entailment cones, and our model. The effect of increasing the embedding dimensions is reported in Appendix B[‡].

We use the validation set to find the best threshold for the classification and obtain the F1 score on the test set using this threshold. Our best performing models have softbox temperature of between 0.2 and 0.5. We obtain the best learning rate for every model using random search and performance on the validation set[§].

## 5. Results

The F1 scores for the hypernymy and meronymy predictions are presented in Table 2. We note that our box embedding method outperforms all baselines by a large margin in the single hierarchy settings. Particularly, in the transitive reduction (0%) setting for hypernymy, we achieve 40% relative improvement compared to the most competitive baseline (order embeddings). This demonstrates the capability of box embeddings to reconstruct the whole transitive closure graph given only the transitive reduction. We observe a similar trend when modeling meronymy. When transitive closure edges are added to the training data (10%, 25%, 50%), the gap between our method and the baselines closes, however, the former still remains superior.

In Table 3, we report the F1 scores for the test edges of the joint hierarchy, which are the composite edges created by the rules (7) mentioned in Section 4.1. We outper-

---

‡. The appendix is available at https://github.com/iesl/Boxes_for_Joint_hierarchy_AKBC_2020/blob/master/supplementary_material/appendix.pdf

§. We use Weights & Biases package to manage our experiments [Biewald, 2020].

| | Hypernym | | | | Meronym | | | |
|---|---|---|---|---|---|---|---|---|
| Transitive Closure Edges | 0% | 10% | 25% | 50% | 0% | 10% | 25% | 50% |
| Order Embedding | 43.0% | 69.7% | 79.4% | 84.1% | 69.7% | 74.1% | 77.3% | 81.0% |
| Poincaré Embedding | 28.9% | 71.4% | 82.0% | 85.3% | 44.7% | 73.6% | 84.9% | 88.0% |
| Hyperbolic Entailment Cones | 32.2% | 85.9% | 91.0% | 94.4% | 49.70% | 83.2% | 88.4% | 92.8% |
| Box Embeddings (w/o regularization) | 45.4% | 72.6% | 81.5% | 89.2% | **83.4%** | 87.2% | 88.7% | 92.6% |
| Box Embeddings (Our Method) | **60.2%** | **90.0%** | **92.7%** | **94.7%** | 80.1% | **91.4%** | **93.8%** | **94.3%** |

Table 2: Test F1 scores of various methods for predicting the transitive closure of WordNet's hypernym meronym relations when training on increasing amounts of edges from the transitive closure. Baselines for hypernymy are as-reported in [Ganea et al., 2018].

| Embedding Model | F1 score |
|---|---|
| Poincaré Embeddings | 43.80% |
| Hyperbolic Entailment Cones | 44.00% |
| TransE | 57% |
| ComplEx | 60.61% |
| Order Embeddings | **68.50%** |
| Box embeddings | 68.10% |

Table 3: Test F1 scores of various methods for predicting the implied HASPART edges

form the multi-relation knowledge base embedding methods as well as hyperbolic models (Poincaré and hyperbolic entailment cones). While methods like TransE and ComplEx lack the inductive bias required to model such highly connected relations, the hyperbolic space methods struggle due to their inability to train very well. Order embeddings, on the other hand, slightly outperform our model, but the latter is essentially at par while also achieving superior performance on the task of modeling single hierarchy. The high performance of order embeddings on this task can be attributed to the "local" nature of the evaluation (we traverse only 2 steps up and down the hypernym graph by keeping $m, n \in \{1, 2\}$ in eq. 7) which might not be exposing the representation limitations of order embeddings – something that is exposed while modeling the relations individually.

### 5.1 Qualitative analysis

In order to visualize the effect of using "soft" boxes, we train a 2-dimensional box embedding model on the hypernym relation of WordNet. Due to the softening of the boxes, the model does not require the *dog.n.01* box to be completely contained inside the *domestic_animal.n.01* box to produce a high score for the edge between them. Figure 5 shows the visualization obtained by plotting the hard edges using the minimum and maximum coordinates of the softboxes. For the Joint Hierarchy model, we obtain similar visualizations which are given in Appendix A[‡].

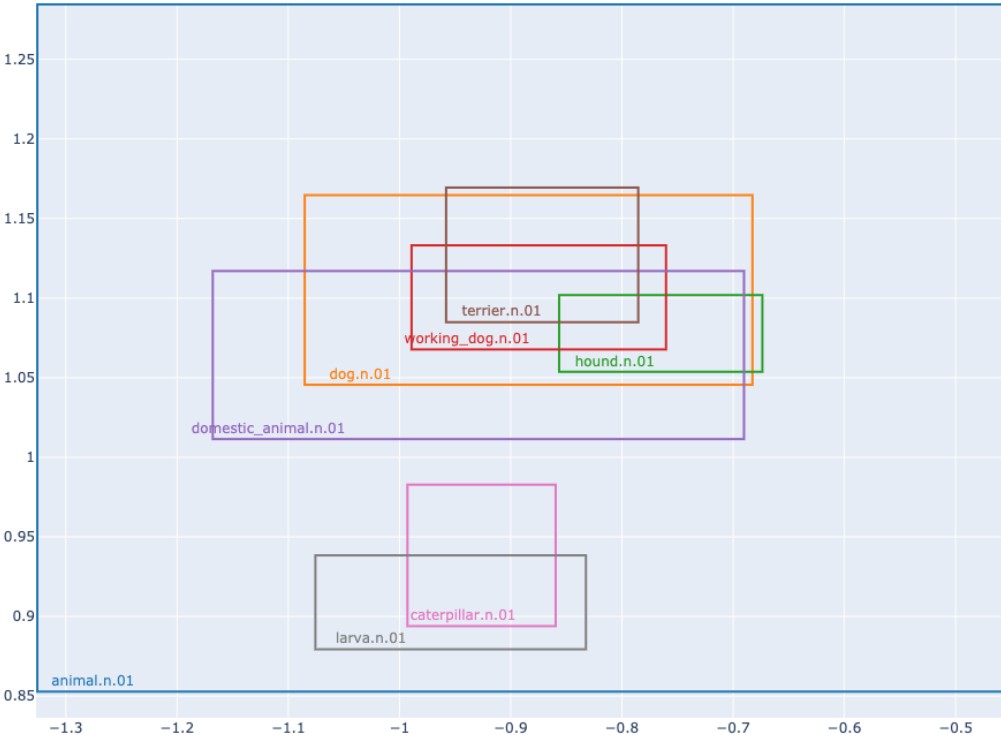

Figure 5: Visualization of 2-dimensional boxes for a subset of entities from the WordNet hypernym hierarchy after training on the entire WordNet reduction.

In order to model a hierarchy, the embedding model has to accommodate the exponential growth in the number of nodes while moving down the hierarchy. The loss function for the box embedding model encourages the embedding of the head node to contain the embedding of the tail node. Hence the number of embeddings having low volume should be high. This intuition is confirmed by Figure 6.

## 6. Conclusion

In this paper we have explored the capability of box embeddings to model hierarchical data. We demonstrated that it provides superior performance compared to alternative methods and requires less of the transitive closure. We also demonstrated that boxes are capable of learning data which is less tree-like, and introduced a method of embedding a joint hierarchy in a single space by augmenting the graph. There are several promising directions which can be explored in future work. It would be of interest to expand the number of relations

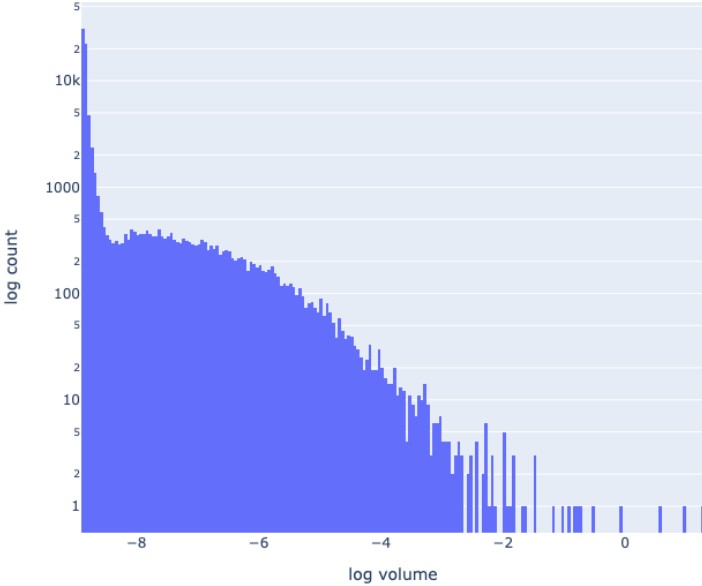

Figure 6: Histogram of the log-volume of 5-dimensional box embeddings for the hypernym relation of WordNet. Here the count (y-axis) is also in log scale.

which are modeled in a single box space and to extend the method to model non-transitive relations. Also, the knowledge encoded in these embeddings can be exploited by using them as a representation layer in neural network models for various downstream tasks like question answering, natural language inference, etc.

## Acknowledgement

This material is based upon work supported in part by the the Chan Zuckerberg Initiative under the project "Scientific Knowledge Base Construction", and in part by the National Science Foundation under Grant No. IIS-1514053. Any opinions, findings and conclusions or recommendations expressed in this material are those of the authors and do not necessarily reflect those of the sponsor. The work reported here was performed in part by the Center for Data Science and the Center for Intelligent Information Retrieval, and in part using high performance computing equipment obtained under a grant from the Collaborative R&D Fund managed by the Massachusetts Technology Collaborative.

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
