# OpenReview forum: "Representing Joint Hierarchies with Box Embeddings"
_AKBC.ws/2020/Conference — AKBC 2020_

### Official Review · AnonReviewer3 · 2020-03-28
**Improving box embeddings but issues with evaluation and little analysis**

**Rating:** 6
**Confidence:** 3

**Review:**

Summary: the paper is elaborating on prior work on Box Embeddings. Box Embeddings represent an object as a rectangle in an n-dimensional space, which makes it possible to represent asymmetric relations (compared with symmetric cosine similarity in regular embeddings). The contributions of the papers are:
- Shows that box embeddings can predict transitive closure from its transitive reduction
- Shows how to use box embeddings (or many other embeddings like order embedding and Poincar´e embedding) to learn multiple edge types jointly in one graph.

Clarity: the paper is easy to read, but it doesn't read like one coherent story. In particular, predicting transitive closure from its transitive reduction (table 2) doesn't fit the main story that focuses on jointly learning multiple relation types (table 3). The reason could be that the results in table 2 are very good, while the results in table 3 are not as good.

Originality: the ideas are not original (mostly Vilnis et al., 2018), but the set of experiments are informative and useful.

Pros:
- Discussing various changes to Box Embeddings training procedure that make then much better than the original work in Vilnis et al., 2018. Most of these changes are prior work though.
- Showing that box embeddings can outperform other baselines by a huge margin when predicting transitive closure from its transitive reduction.
- A simple and general method to convert graphs with multiple edge types into graphs that can be learned using box embeddings and other embedding methods.

Cons:
- Isn't predicting transitive closure from its transitive reduction a procedural operation that doesn't require a trained model? The evaluation setup of Nickel and Kiela, 2017, Ganea et al., 2018 drops random edges, which means it can run into situations where a trained model is needed to fill in missing evidence.
- No analysis to explain why box embeddings does so well in predicting transitive closure compared to other embedding methods, and why it doesn't do as good in predicting multiple relation types (comparable to Order Embeddings)

Notes and questions:
- How did you choose the embedding size? do you think the results are agnostic to it?
- #nodes = #entities^#relation_types, which means it will explode quickly as the relation types increase. Is there a way around that?
- I didn't get why you need to double the number of nodes to represent both relation types. Would it be easier to use ternary (instead of binary) class classification (not related, hypernymy, meronymy)?

---

> ### Author Response · Authors · 2020-04-09
> **Response to Reviewer 3**
>
> We thank the reviewer for detail and constructive feedback.
>
> We address the comments and concerns of the review below:
>
> RE: Clarity
>
> We agree that it might seem that the two points made in the paper are weakly connected.
> However, we observed that modeling a joint hierarchy requires much of the same inductive bias that is essential to model a transitive relation like the hypernym hierarchy. Hence, we begin our experiments by evaluating the modeling capacity of different methods on the individual hierarchies and finally, we evaluate those same methods on the task of modeling the joint hierarchy.
>
> RE: Evaluation protocol
>
>  The task here is to model hierarchical data such that the transitive closure edges can be readily implied from the transitive reduction edges without having to run any inference algorithm. Such representations embedded in a continuous space can be used as a representation layer for a neural network based model on a downstream task. Hence, we want to test the modeling capacity of the models and not their generalization power. While this is our motivation for using this evaluation setup, it turns out that “Ganea et al., 2018” follow the exact same setup. That is, all the basic edges (transitive reduction edges) are present in their training set, and the edges in the dev and test are produced by randomly sampling from the set of non-basic edges (implied transitive closure).
>
>  RE: Regarding the analysis of performance on modeling the joint hierarchy
>
>  While all the models, including the baselines, get improved performance as we add more transitive closure edges to the training set, we hypothesis that there could be two reasons for the improvement. First, adding transitive closure edges in the training nullifies the effect of them appearing in the random negatives (as bad negatives). Second, these edges convey the information about the depth of a particular node and hence help the model learn better. We believe that the training of Poincare embeddings and hyperbolic entailment cones relies heavily on the availability of the depth information (ie transitive closure edges). While the improvements for our box embedding method and for the order embedding method can majorly be attributed to the first point mentioned above. That being said, it is not clear how to measure the effects of the two points mentioned.
>
>  Regarding the performance on joint hierarchy, it was surprising that order embeddings were slightly better than box embeddings here and it is not entirely clear why. One possibility is that the joint hierarchy task has a more “local” evaluation (where we traversed 2 steps up and down the hypernym graph) and the representation limitations of order embeddings are not exposed in this evaluation.
>
>
>  RE: Embedding size
>
> Reviewer 2 had a similar question and the factors leading to picking the reported embedding sizes are detailed in the response (RE: The use of small embedding dimensions). As seen in the table in that response, all the models improve as the embedding dimension increases but the box embeddings are still better in all the cases.
>
>  RE: Number of parameters exploding with increasing number of relations
>
> Please note that we have #nodes = #entities*#relation_types with our method. That being said, your point is still important — number of boxes does increase if we add more relations which have the semantics of joint hierarchy. While currently, we do not have a way around, we are trying the develop methods which use transformations to share weight parameters across different relations to limit the increase in parameters.
>
>  RE: Converting the problem to ternary classification
>
>  In general, if there is absolutely no parameter/information sharing between the relations, given that we have two relations, one should expect to have one embedding per entity per relation. Each of these relation-wise embeddings would be in their own space.  In this work, we attempt to share information between the two hierarchical relations by embedding them in the same space. However, according to us, the best way to exploit the fact that both the relations are hierarchical is to have two embeddings per node. As described in section 2.5, and in figure 4.
>
> We do not use ternary classification here because we are trying to learn representations that readily answer the question, “Does an edge of (a given) relation R exists between the given two entities?”
>
> If this answer is not sufficient, could you please elaborate a little bit more about this particular question.

---

### Official Review · AnonReviewer1 · 2020-03-28
**Interesting idea, reasonable results, but might have some flaws**

**Rating:** 6
**Confidence:** 3

**Review:**

This paper studies the properties of box embedding from three aspects: 1) how many edges from the transitive closure are necessary for training the box embeddings; 2) can box embeddings be applied to graphs that are not tree-like; 3) how to model different relations in the same space. The author set up experiments using the IsA relation (Hypernymy) and HasPART relation (Meronymy)  in WordNet. The results show the effectiveness of box embedding, outperforming other embedding method including order embedding, Poincar´e embedding, hyperbolic entailment cones, TransE and ComplEx.

I am inclined to accept this paper because:

* The study explores interesting topic, and show superior properties of box embedding.
* They propose a novel approach to jointly modeling different relations in the same box embedding space.

But I also have several concerns, which I want the author to address:

* The embedding dimension is 10 for baselines, and 5 for your model. They seem too small compared to the embedding methods I am familiar with, and make the box embedding more like a toy model. Is it standard in previous work of box embeddings?
* I am wondering what is the difference between your model and previous box embedding methods when you just need to model the hypernym relations. Also, can you also compare previous box embedding methods as baselines in Table 2 and Table 3?
* The title seems to focus on modeling the joint hierarchies, but this is only one point explored in the paper. Also, the proposed method actually didn't outperform order embedding method in the this joint hierarchy task. It would be better if you have some discussion in the paper on why order embedding becomes much better here.
* For the visualization in Figure 5, why do you only use the hypernym hierarchy? My suggestion is to include all the three hierarchies here.

---

> ### Author Response · Authors · 2020-04-09
> **Response to Reviewer 2**
>
> We thank the reviewer for detailed and constructive feedback.
>
> We address the reviewer’s concerns below:
>
> RE: The use of small embedding dimensions
>
> We keep the embedding dimension 5 (10 parameters) to compare our method to the results of the baselines with dimension 10 (as reported in Ganea et. al., 2018).  All these embedding methods, including ours, have a strong inductive bias towards modeling the hierarchical relations and thus requires significantly less number of dimensions to model the type of dataset that is being used. We did some additional experiments to see how different baselines and box embeddings scale with increasing number of dimensions for the WordNet Hypernym dataset. Following table shows our findings for the hypernym relation with 0% transitive closure edges in the training set (we will include this experiment in the appendix of the revised version of the paper)
>
>     | Model                                                                    | Test F1 score                    |
>     |------------------------------------- -------------------- |---|------|------|------|------     |
>     | Vector Dimension (parameters per node) |   | 10   | 20   | 50   | 100      |
>     | Order Embedding                                           |   | 43   | 49.2 | 52.2 | 53     |
>     | Poincare                                                            |   | 28.9 | 30.9 | 31.2 | 31.4 |
>     | Hyperbolic entailment cones                        |   | 32.2 | 33.2 | 38.5 | 39.4 |
>     | Box Embedding (our method)                      |   | 60.2 | 65.0 | 67.0| 68.0   |
>
> We observe that all of the methods get similar performance boost up to dimension 50, after that, their increase in performance plateaus. We hypothesis that the plateau in the performance of box embeddings on this dataset is due to noisy negative sampling and it can be improved with intelligent negative sampling strategies.
>
> Previous work on box embeddings [Li et al., 2019] shows that as the dataset size grows, box embeddings can take advantage of increasing dimensions and continue to improve its performance. This is evident from the fact that [Li et al., 2019]  used 300 dimensions for a large entailment dataset like Flickr. Hence, even though it is not a standard in the previous work on box embeddings, in this work, to compare with the chosen baselines [Nickel and Kiela, 2017, Ganea et. al., 2018] , we keep the number of dimensions low.
>
> RE: Difference between the previous Box embeddings and our method when modeling hypernymy
>
> The embedding method used here is indeed very similar to the previous work (Li et al., 2019) on box embeddings. However, the previous box embedding models do not use the regularization term that we propose in this paper.  We observe that this regularization is really important and gives a significant performance boost. For instance, for the task of modeling hypernymy (Table 2, 0% transitive closure edges in the training set), our method improves upon the previous box embedding method (Li et al., 2019) by around 17 F1 points. We are currently running the experiments for other settings (10%-50% of transitive closure edges) and will include that in Table 2 in the revised version of the paper.
>
> RE: The title and the performance on the task of modeling joint hierarchy
>
> Indeed, we were surprised that order embeddings were slightly better than box embeddings here and it is not entirely clear why. One possibility is that the joint hierarchy task has a more “local” evaluation (where we traversed 2 steps up and down the hypernym graph) and the representation limitations of order embeddings are not exposed in this evaluation. We would point out, however, that box embeddings are essentially “at par” with order embeddings on this evaluation, and vastly outperform order embeddings on the single-relation evaluations. We will include additional discussion of this point in the revision of the paper.
>
>
>  RE: Visualizations of Joint Hierarchy
>
>  We have generated visualizations for the model trained on the joint hierarchy and will be including those in the appendix of the revised version of the paper.

---

### Official Review · AnonReviewer2 · 2020-03-30
**Review: Representing Joint Hierarchies with Box Embeddings**

**Rating:** 8
**Confidence:** 3

**Review:**

The authors have explored the capability of box embeddings to model hierarchical data, and have demonstrated that it provides superior performance compared to alternative methods, requiring less of the transitive closure. In terms of F1 score for measuring quality of hypernym and meronym predictions, the authors find that their box embedding method outperforms all baselines by a large margin in the single hierarchy settings.

Strong points of the paper:

--Reasonably well written and rigorously presented

--Notwithstanding the sole use of WordNet, the baselines and experimentation left a good impression. The authors were reasonably thorough.
--The writing was good, but I would have liked to see an explicit formulation of the binary cross-entropy or the regularized loss that the authors were minimizing for the sake of completeness. From what I can see, the expression has to be derived based on what the authors have written in the text.


Weak points:

Perhaps the most significant weakness is the exclusive use of WordNet for demonstrating effectiveness. Either a supporting dataset (e.g., in the context of a task like commonsense question answering) or another knowledge base would have lent stronger credence to the claims.

Introduction was a mixture of a true introduction and related work. I think the authors should have kept their introduction at a higher level

The last plot in the paper could have been log-log to show the trends more clearly.

---

> ### Author Response · Authors · 2020-04-09
> **Response to Reviewer 1**
>
> We thank the reviewer for the positive remarks and valuable suggestions.
>
> We address the concerns below:
>
> RE: Exclusive use of WordNet for demonstrating effectiveness
> This is a valid concern. However, in this paper, we wish to demonstrate the capacity of the box embeddings to capture hierarchies and also joint hierarchies.  The next step is to use such a model as a representation layer for downstream tasks like common-sense question answering. However, the purpose of this work is to limit the evaluation to measuring the modeling capacity and then pick the best representation model to perform a downstream task in future work. Since WordNet’s well defined hierarchical relations allow us to test the modeling capacity rigorously, we stick to using WordNet to evaluate as many baseline models as possible. We are also exploring the possibility of applying box embeddings as a representation layer for the task of commonsense based subject-verb-object prediction as future work.
>
> RE: Reformatting the Introduction.
> We are working on rewriting the introduction and separating out the related work from it. We will upload a revised version with these changes before the due date.
>
> RE: log-log plot to capture the trend better.
> We have generated (will be included in the revised version) a new log-log plot which indeed captures the distribution of volume more clearly.

---

### Decision · Program_Chairs · 2020-05-01

**Decision:**

Accept

**Comment:**

Reviewers unanimously appreciated this paper. Please do take into account their feedback to improve the paper.
From our perspective, the paper is not written in a scholarly fashion: there is so much work on hierarchical models, learning embeddings of trees, and why not give credit to these people? Please expand your related work discussion and give proper context.